# Synthetically modified guide RNA and donor DNA are a versatile platform for CRISPR-Cas9 engineering

Kunwoo Lee[1]*, Vanessa A Mackley[1], Anirudh Rao[2], Anthony T Chong[2], Mark A Dewitt[3,4], Jacob E Corn[3,4], Niren Murthy[2]*

[1]GenEdit Inc, Berkeley, United States; [2]Department of Bioengineering, University of California, Berkeley, Berkeley, United States; [3]Department of Molecular and Cell Biology, University of California, Berkeley, Berkeley, United States; [4]Innovative Genomics Institute, University of California, Berkeley, Berkeley, United States

**Abstract** Chemical modification of the gRNA and donor DNA has great potential for improving the gene editing efficiency of Cas9 and Cpf1, but has not been investigated extensively. In this report, we demonstrate that the gRNAs of Cas9 and Cpf1, and donor DNA can be chemically modified at their terminal positions without losing activity. Moreover, we show that 5' fluorescently labeled donor DNA can be used as a marker to enrich HDR edited cells by a factor of two through cell sorting. In addition, we demonstrate that the gRNA and donor DNA can be directly conjugated together into one molecule, and show that this gRNA-donor DNA conjugate is three times better at transfecting cells and inducing HDR, with cationic polymers, than unconjugated gRNA and donor DNA. The tolerance of the gRNA and donor DNA to chemical modifications has the potential to enable new strategies for genome engineering.

*For correspondence: lee@ genedit.com (KL); nmurthy@ berkeley.edu (NM)

## Introduction

The CRISPR/Cas9 system uses a gRNA to target and cleave DNA sequences with specificity, and can result in precise genome editing via homology directed repair (HDR) if a donor DNA template is simultaneously delivered along with the CRISPR/Cas9 system (*Jinek et al., 2012*; *Cong et al., 2013*; *Cho et al., 2013*; *Mali et al., 2013a*; *DeWitt et al., 2016*; *Long et al., 2014*). Modifications to the gRNA and the donor DNA are effective methods to engineer CRISPR/Cas9 to enhance editing and develop new applications. For example, extension of the gRNA with aptamer sequences can be used to recruit RNA binding counterparts within the cell that enable either the visualization or transcription of specific DNA sequences in the genome (*Mali et al., 2013b*; *Chen et al., 2013*). In addition, incorporation of modified RNA bases into the gRNA results in reduced immunogenicity, increased stability, and enhanced gene editing efficiency (*Rahdar et al., 2015*; *Hendel et al., 2015*). Finally, chemical modification of the donor DNA with phosphorothioates dramatically increases HDR efficiency by increasing donor DNA stability (*Renaud et al., 2016*). Thus, modification of either the guide RNA or the donor DNA has great potential for enhancing the biotechnological applications of the CRISPR/Cas9 system.

However, outside of base modifications and backbone modifications, very little is known about the chemical modifications that are tolerated by the gRNA of the CRISRP/Cas9 system and the donor DNA. In particular, it is uncertain if the gRNA of Cas9 and the donor DNA tolerate modifications with chemical moieties that are structurally unrelated to nucleotide bases, and how big these modifications can be without affecting the functionality of Cas9 or donor DNA. In addition, nothing is known about the tolerance of the Cpf1 gRNA to chemical modifications. Information about the

**eLife digest** There are several different technologies that can be used to make specific changes to particular genes in cells. These "gene editing" approaches have the potential to help humans in a variety of different ways, for example, to treat diseases that presently have no cure, or to improve the nutritional quality of crop plants. One such gene editing approach is known as CRISPR. To edit a specific gene, a molecule called a guide ribonucleic acid (or guide RNA for short) binds to a section of the gene and recruits an enzyme to cut the DNA encoding the gene in a particular location. Adding a "donor" DNA molecule that contains the desired "edit" can lead to the cell repairing the broken gene in a way that incorporates the desired change.

Modifying the guide RNA or the donor DNA can enhance CRISPR editing. For example, extending the guide RNA molecules by adding "aptamer" sequences can enable researchers to specifically activate the genes that have been edited. It is also possible to add chemical tags to RNA and DNA, but it is not clear how chemical modifications to the guide RNA and donor DNA could affect CRISPR.

Here Lee et al. investigated whether adding chemical tags to the guide RNA and/or donor DNA could enhance gene editing. The experiments show that the modified guide RNAs and donor DNAs were still active and could edit DNA in mouse and human cells. Adding a fluorescent molecule to the donor DNA allowed Lee et al. to track which cells contained donor DNA and separate them from other cells. The fluorescent cells had twice as much editing compared to groups of unsorted cells.

In further experiments, the guide RNA and donor DNA were fused together and supplied to cells together with a DNA cutting enzyme. Cells containing this combined molecule had three times more editing than cells exposed to the original CRISPR system. This change may aid the development of new uses for CRISPR because it simplifies the system from three components (an enzyme, guide RNA and donor DNA) to just a cutting enzyme and the combined molecule.

Overall, the findings of Lee et al. show that chemical modifications to guide RNA and donor DNA can make the CRISPR system more versatile. It opens up the possibility of new applications such as adding a targeting group that would direct the CRISPR Cas9 system to a specific cell type or tissue.

chemical modifications tolerated by the gRNA and donor DNA can provide new ways to engineer the CRISPR/Cas9 system for high efficiency editing and also enable new applications of the CRISPR/Cas9 system not related to DNA cleavage.

In this report, we investigated the types of synthetic modifications that the gRNA and donor DNA can tolerate in the CRISPR gene editing systems, and demonstrate that the gRNA and the donor DNA can be modified at their terminal positions with large modifications ranging from planar hydrophobic molecules to 35 kDa nucleic acids, without losing activity. We exploited the tolerance of the donor DNA to 5' modifications to develop a method for enriching cells that had been edited via HDR (*Figure 1a*). In addition, we also exploited the tolerance of the donor DNA and gRNA to terminal modifications to develop a method for increasing the delivery and HDR efficiency of the CRISPR/Cas9 system and donor DNA in cells (*Figure 1b*).

## Results and discussions

### The Cas9 and Cpf1 gRNAs tolerate large chemical modifications at their terminal positions

Very little is known about the tolerance of the gRNAs of Cas9 and Cpf1 towards chemical modifications. Without this information, it is challenging to rationally engineer gRNAs for biotechnological applications. We therefore generated a small library of 8 chemically modified CRISPR targeting RNAs (crRNAs), which had modifications at their 5' or 3' ends, and evaluated their ability to cleave genomic DNA, after complexation with Cas9, in cells expressing blue fluorescent protein (BFP). The chemical modifications we synthesized or purchased are shown in *Figure 2a*. The library consisted of crRNAs targeting the BFP sequence, which had an amine, azide, fluorescent dye, strained alkyne,

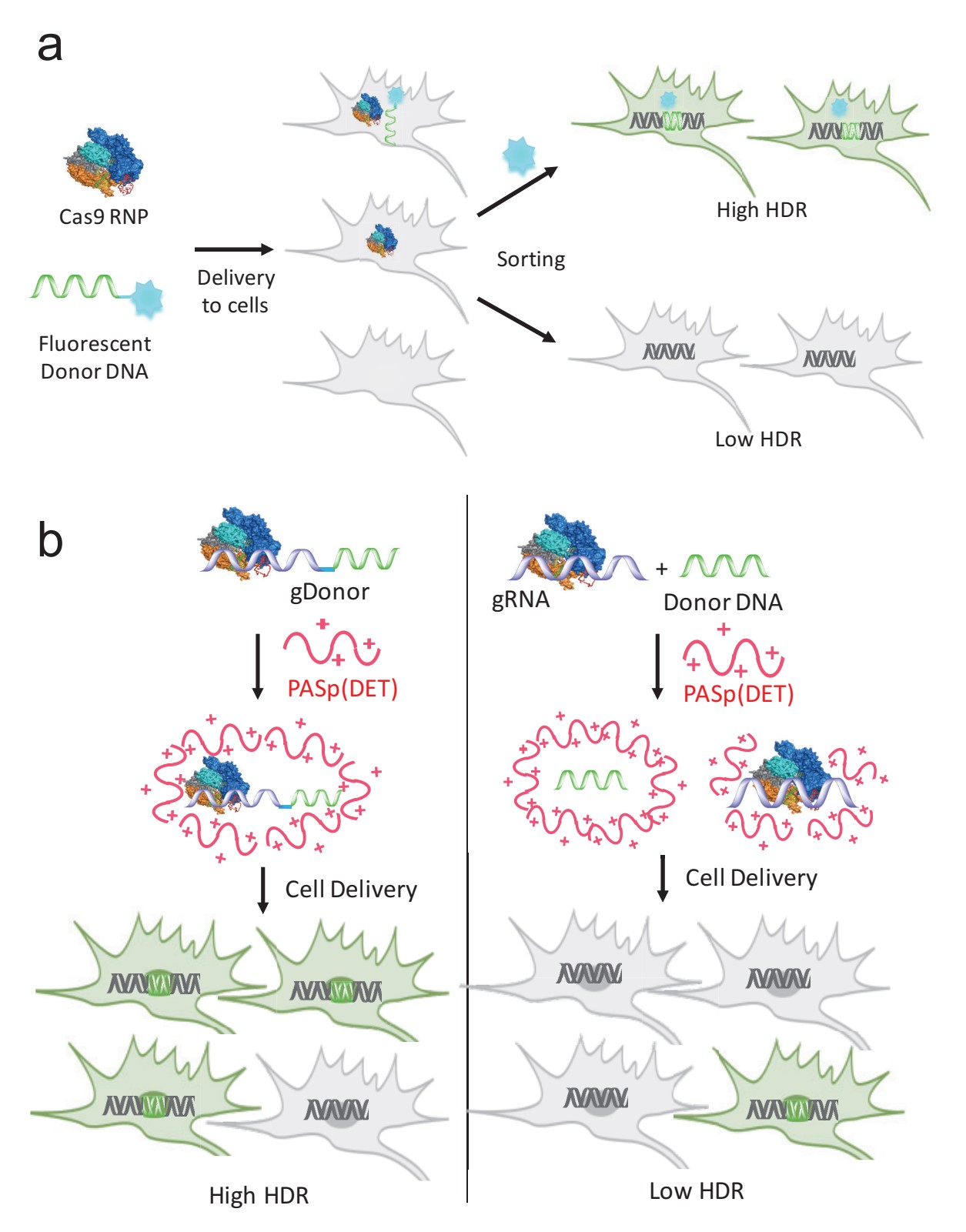

**Figure 1.** gRNA and donor DNA engineering enables the development of new strategies for enriching gene edited cells and for improving their delivery into cells. (a) Fluorescently labeled donor DNA can be used as a marker for rapidly enriching HDR edited cells. Cells that internalize fluorescent donor DNA have a high probability of being gene edited via HDR, and can be isolated via FACS based on fluorescence intensity. (b) The gRNA and donor DNA can be conjugated together to generate a single molecule (gDonor). Cas9 complexed to gDonor is more efficient at inducing HDR in cells,

*Figure 1 continued on next page*

*Figure 1 continued*

after transfection with cationic polymers, than free gRNA complexed to Cas9 and donor DNA. The gDonor/Cas9 complex binds polycations, and the resulting nanoparticles have both gRNA and donor DNA in a single nanoparticle, leading to efficient HDR in cells (left). In contrast, Cas9 RNP + donor DNA forms heterogenous complexes with polycations, which are unable to encapsulate Cas9 RNP and donor within the same nanoparticle, leading to low efficiency HDR in cells (right).

The following figure supplements are available for figure 1:

**Figure supplement 1.** Synthesis of DBCO-crRNA.

**Figure supplement 2.** Synthesis of DNA-crRNA.

**Figure supplement 3.** The synthesis of DNA-crRNA was confirmed with gel electrophoresis.

disulfide, and a short single stranded DNA (87 nucleotides), at their 5' or 3' position. These modifications were chosen because of their importance in performing conjugation reactions.

crRNAs with the chemical modifications shown in *Figure 2a* were complexed with tracrRNA and Cas9 and electroporated into BFP-HEK cells. The percentage of BFP negative cells was determined via flow cytometry 5 days after the electroporation. *Figure 2b* demonstrates that the crRNA for Cas9 tolerates large modifications at its 5' end, and is less tolerant to modifications on the 3' end. For example, 5' modified crRNAs had a similar non-homologous end joining (NHEJ) frequency in BFP-HEK and BFP-K562 cells (*Figure 2b* and *Figure 2—figure supplement 1*) as control unmodified crRNA. In contrast, crRNA with 3' modifications had a 50% reduction in NHEJ efficiency in cells. The sensitivity of crRNA to 3' modifications is anticipated as the 3' of the crRNA in Cas9 hybridizes with tracrRNA and is in close proximity to the Cas9 protein, whereas the 5' end has a looser interaction with Cas9 (*Jiang et al., 2015*; *Yamano et al., 2016*; *Jinek et al., 2014*; *Fu et al., 2014*).

We also investigated the tolerance of the Cpf1 guide RNA to chemical modifications. Cpf1 is a recently discovered RNA-guided endonuclease of the class 2 CRISPR-Cas, and has the potential to be an alternative to Cas9 as it recognizes non-classical PAM sequences (*Yamano et al., 2016*; *Zetsche et al., 2015*; *Kleinstiver et al., 2016*; *Zetsche et al., 2017*). Unlike Cas9, which requires both crRNA and trans-activating crRNA (tracrRNA), Cpf1 requires only crRNA, and therefore it is an even more attractive target for chemical modifications. crRNA targeting the BFP gene and Cpf1 were electroporated into BFP-HEK cells, and the percentage of BFP negative cells was quantified by flow cytometry. *Figure 2c* demonstrates that the crRNA from AsCpf1 (from Acidaminococcus) tolerates chemical modifications such as amine, azide, and DBCO at its 3' end; in contrast, bulky 5' end modifications such as azide and DBCO are less tolerated. For example, BFP-HEK cells electroporated with 3' amine-crRNA and Cpf1 had a similar NHEJ frequency as cells electroporated with Cpf1 and unmodified crRNA. In contrast, BFP-HEK cells electroporated with crRNAs with 5' modifications and Cpf1 had a reduction in NHEJ frequency and generated only 50–80% of the NHEJ levels as cells treated with unmodified crRNA. The 5' nucleotide of the crRNA has more interaction points with the Cpf1 protein than the 3' nucleotide, and the sensitivity of the 5' end to chemical modifications corresponds with the crystal structure of Cpf1 (*Yamano et al., 2016*). Thus, the gRNAs for Cas9 and Cpf1 can be modified at one of their terminals with minimal loss of activity, and this tolerance should enable a variety of new strategies for gRNA engineering.

## The donor template DNA tolerates chemical modifications

In addition, we also investigated the tolerance of the donor DNA to chemical modifications. We investigated the HDR efficiency of donor DNA with 5' and 3' modifications composed of an azide modification, an amine modification, and an Alexa 647 modification (*Figure 2d*). For these experiments, a donor DNA which converts the BFP gene to the GFP gene was electroporated in cells along with Cas9 RNP targeting the BFP gene (*Richardson et al., 2016*). *Figure 2e* demonstrates that the donor DNA tolerates modifications at both its 5' and 3' ends, and BFP-HEK cells electroporated with chemically modified donor DNA were efficiently converted to GFP expressing cells via HDR. The tolerance of the donor DNA to chemical modifications at the 5' and 3' ends is anticipated based on the role of the donor DNA in the HDR mechanism. The donor DNA is rarely incorporated

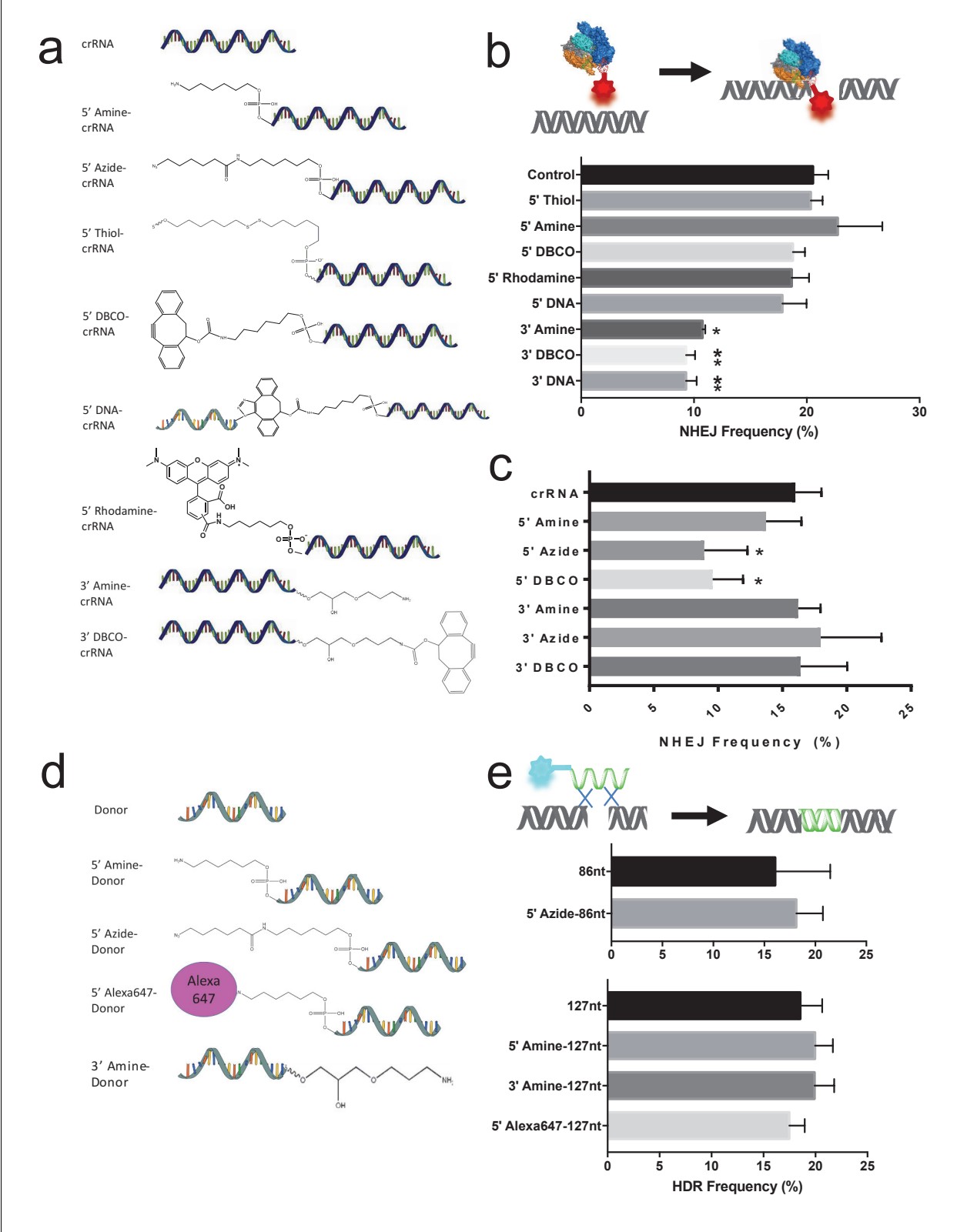

**Figure 2.** The gRNAs for Cas9 and Cpf1 and donor DNA tolerate large chemical modifications at their terminal ends. (a) Chemical structure of modified gRNAs. gRNAs with 5' or 3' modifications were purchased or synthesized. (b) Cas9 crRNAs with 5' or 3' modifications and Cas9 were electroporated into BFP-HEK cells, and their activity was quantified by determining the amount of NHEJ they generated in cells (% BFP negative cells). crRNAs tolerate 5' modifications but are sensitive to modifications at the 3' end. DNA-crRNAs are crRNAs conjugated to an 87nt scrambled DNA oligonucleotide. One

*Figure 2 continued on next page*

## eLIFE Short report

Genes and Chromosomes | Human Biology and Medicine

*Figure 2 continued*

way ANOVA, post-hoc Tukey test, significant difference from control, *$p<0.05$, **$p<0.01$. (c) Cpf1 crRNAs with 5' or 3' modifications and Cpf1 were electroporated into BFP-HEK cells, and their ability to generate NHEJ was investigated. One way ANOVA, post-hoc Tukey test, significant difference from control crRNA, *$p<0.05$. (d) Chemical structures of modified donor DNA. Donor DNA with 5' or 3' modifications were purchased or synthesized. (e) Donor DNA with 5' or 3' modifications and Cas9 RNP were electroporated into BFP-HEK cells and their ability to induce HDR was investigated. The donor DNA tolerates both 5' and 3' modifications.

The following figure supplement is available for figure 2:

**Figure supplement 1.** The NHEJ frequency of BFP-K562 cells transfected with chemically modified crRNAs.

directly into the genome after a double stranded break, and instead, acts as a template for the DNA repair. Therefore, modifications of the 5' and 3' ends are likely to be tolerated (*Holloman, 2011*; *Durai et al., 2005*). Thus, the donor DNA can be modified at its 5 or 3' position without losing activity, and this tolerance should enable a variety of new strategies for donor DNA engineering.

## Fluorescently labeled donor DNA can be used to enrich for cells that are likely to be edited via HDR

Chemically modified gRNAs and donor DNA can potentially be used to address technological challenges limiting progress in the field of genome editing, such as the lack of methods available for rapidly identifying cells genetically edited via HDR. Current methods for identifying gene edited cells requires deep sequencing of randomly selected clones or the introduction of new additional proteins for magnetic bead pull down or fluorescence labeling (*Lee et al., 2016*; *Kim et al., 2013*). However, neither of these methods is ideal for selecting gene edited cells and alternatives are greatly needed. For example, randomly choosing clones from cell transfection experiments and sequencing them to identify gene edited cells requires significant levels of investment and requires several weeks to complete. Similarly, introducing additional proteins into cells for the purpose of purification causes numerous additional complications, such as cell perturbation and unnecessary foreign protein expression. In addition, most cell identification and enrichment methods take several days to complete, which are too long for many primary cells because they rapidly change their phenotype after culturing *in vitro*. The inability to rapidly identify gene edited cells currently limits the development of gene edited cell therapies and the development of engineered cell lines (*Dever et al., 2016*). Therefore, the development of a fast and non-invasive method for enriching gene-edited cells has the potential to accelerate progress in multiple areas of genome engineering.

The donor DNA concentration in the nucleus is a key factor that determines the HDR efficiency after a double stranded break. We therefore investigated if donor DNA that was fluorescently labeled could be used as a beacon for FACs sorting, to identify cells that had a high probability of undergoing HDR, after transfection with Cas9 RNP (*Figure 1a*). Enriching cells for gene editing, non-invasively via FACS, has the potential to significantly accelerate gene editing cell culture protocols, because it will reduce the number of clones that need to be isolated and sequenced. A donor DNA was labeled with Alexa 647, termed trackable Donor (tDonor), and was electroporated into BFP-HEK cells along with Cas9 RNP. Sixteen hours after the electroporation, cells that internalized high levels of the tDonor and low levels of the tDonor were sorted using fluorescence activated cell sorting (FACS). After three days of culture, the HDR frequency was determined using flow cytometry and compared to bulk unsorted cells. *Figure 3* demonstrates that tDonor can be used as a selectable marker to enrich for cells that have a high probability of being edited via HDR. Cells that had internalized high levels of the donor DNA had a high rate of HDR, and the HDR rate in these cells increased by a factor of 2. Likewise, sorted cells that had low levels of tDonor showed significantly lower levels of HDR editing, demonstrating that the amount of Donor DNA in a cell is an important factor for HDR (*Figures 3b, c and d*).

In addition, we investigated if sorting cells based on the amount of donor DNA internalized would help in identifying primary cells that had been edited via HDR. Primary myoblasts from the Duchenne muscular dystrophy mouse model (mdx mice), which have a mutation in their dystrophin gene, were used for these experiments Primary myoblasts were transfected with Cas9 RNP and a fluorescently labeled tDonor designed to correct the dystrophin mutation, using lipofectamine. The transfected

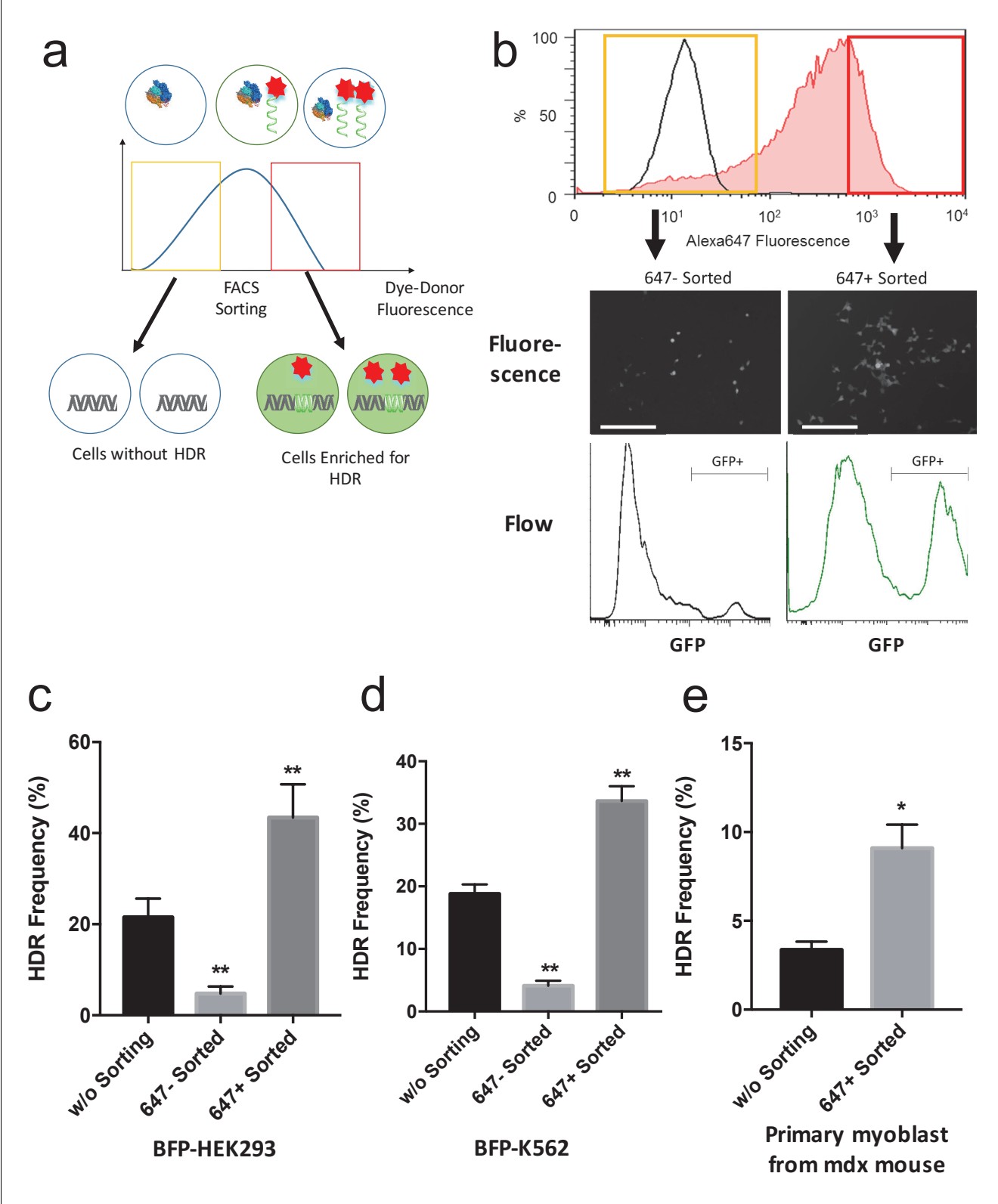

**Figure 3.** Fluorescently labeled donor DNA can be used to enrich for cells that have been edited via HDR. (**a**) Overview of the cell enrichment process. Cells are transfected with Alexa647-donor DNA (tDonor) and Cas9 RNP, and sorted based on their intracellular levels of Alexa647-donor DNA via FACS. Cells with high levels of Alexa647-donor DNA are enriched for HDR. (**b**) BFP-HEK cells were electroporated with Cas9 RNP and Alexa647-Donor, and were sorted via FACS based on Alexa647 fluorescence. The histogram shows the FACS analysis of Alexa647 fluorescence, control untreated cells are in

*Figure 3 continued on next page*

*Figure 3 continued*

black, and cells electroporated are in red. The yellow box shows the gating used to identify the Alexa 647 negative cells (bottom 20% gating), and the red box shows the gating used to identify the Alexa 647 positive cells (top 20% gating). Fluorescent images and histograms from flow cytometry of the sorted cells demonstrates that cells with high amounts of Alexa647-Donor had higher levels of HDR (bar: 100 μm). (c) The HDR rate in BFP-HEK cells was determined by quantifying GFP expression. The bulk population of transfected cells (without sorting), cells with low levels of Alexa647-Donor, and cells with high levels of Alexa647-Donor were analyzed by flow cytometry. Alexa647 based sorting enriches for cells that have a high probability of being edited via HDR. One way ANOVA, post-hoc Tukey test, significant difference from control, *p<0.05, **p<0.01. (d) The HDR rate in BFP-K562 cells was determined by quantifying GFP expression. The bulk population of transfected cells (without sorting), cells with low levels of Alexa647-Donor, and cells with high levels of Alexa647-Donor were analyzed by flow cytometry. Alexa647 based sorting enriches for cells that have a high probability of being edited via HDR. One way ANOVA, post-hoc Tukey test, significant difference from control, *p<0.05, **p<0.01. (e) Primary myoblasts from mdx mice were transfected with Cas9 RNP and Alexa647-Donor using lipofectamine, and were sorted via flow cytometry based on Alexa647 fluorescence. The correction of the dystrophin mutation in these cells via HDR was quantified by restriction enzyme analysis of the dystrophin gene. Flow sorted cells that internalized high amounts of the Alexa647 donor had more than a 2-fold increase in HDR frequency than unsorted cells. One way ANOVA, post-hoc Tukey test, significant difference from control, *p<0.05, **p<0.01.

The following figure supplement is available for figure 3:

**Figure supplement 1.** BFP-K562 cells with high levels of fluorescently labeled donor DNA are enriched for HDR edited cells.

cells were sorted via flow cytometry, using the fluorescence of the tDonor for gating, cultured, and analyzed for gene editing via restriction enzyme analysis. *Figure 3e* demonstrates that the HDR rate in primary myoblasts with high levels of tDonor is two fold higher than unsorted cells. Fluorescently labeled donor DNA represents an easy and fast method for enriching gene edited cells, and could find numerous applications in biology and medicine.

## A gRNA-donor DNA conjugate (gDonor) can function as a gRNA and a donor DNA

There is great interest in developing therapeutics based on the CRISPR/Cas9 system; however, delivery problems have limited their clinical progress. Although viral vectors based on AAV can efficiently deliver Cas9 (*Long et al., 2016*; *Tabebordbar et al., 2016*; *Nelson et al., 2016*), their toxicity is problematic. Therefore there is great interest in developing non-viral delivery vehicles that can deliver Cas9, gRNA and donor DNA into cells (*Yin et al., 2016*). Cationic polymers are a promising delivery vehicle for Cas9, as they avoid several of the problems associated with using AAV, such as immunogenicity and off-target DNA damage caused from the sustained expression of Cas9. However, simultaneous delivery of donor DNA and Cas9 RNP into cells is challenging with cationic vectors, because the donor DNA and the Cas9 RNP have different charge densities, and it is therefore unlikely that a single nanoparticle will contain both Cas9 RNP and donor DNA. As a result, it is challenging to deliver Cas9 RNP and donor DNA simultaneously into the same cell and efficiently induce HDR.

To circumvent this obstacle, we performed experiments to determine if the gRNA and donor DNA could be combined into a single molecule (termed gDonor). gDonor should complex polycations even after binding the Cas9 protein because the gDonor has a much higher charge density than either the gRNA or donor DNA, due to its longer length (*Figure 4a*). In addition, gDonor should significantly increase the transfection efficiency of polycationic vectors because every cell that has internalized gRNA will have also have internalized donor DNA. Finally, using gDonor for inducing HDR ensures that there is donor DNA in the vicinity of the DNA cleavage site, and this may further promote efficient HDR (*Holloman, 2011*; *Durai et al., 2005*).

A gRNA-donor DNA conjugate (gDonor) was synthesized by conjugating an azide terminated donor DNA with an alkyne modified crRNA, and hybridizing the resulting conjugate with tracrRNA. The gRNA was designed to cut the BFP gene and the donor DNA was designed to convert the BFP gene into the GFP gene. The gDonor was purified via gel extraction, and was synthesized with a 40% yield (*Figure 4b*). The activity of the gDonor was investigated by determining its ability to induce NHEJ or HDR in BFP-HEK cells, after electroporation with the Cas9 RNP. In addition, the DNA cleavage pattern of the gDonor in cells was also compared against cells treated with Cas9 RNP and donor DNA to determine whether conjugation to the donor DNA affected the function of the

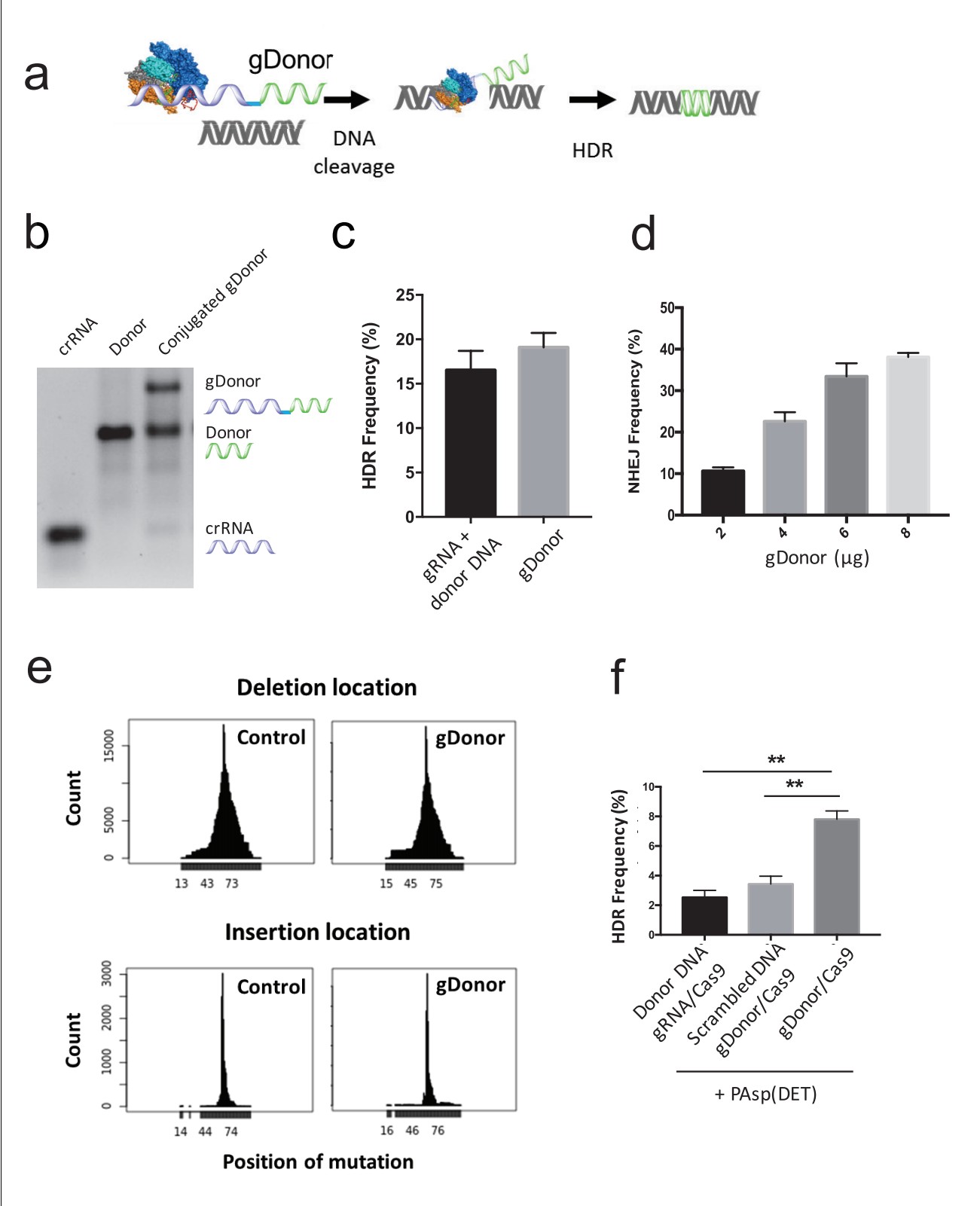

**Figure 4.** A gRNA-donor DNA conjugate (gDonor) transfects cells with higher efficiency than free gRNA and donor DNA. (**a**) The proposed mechanism of gene editing with gDonor/Cas9 complexes in cells. (**b**) Synthesis of gDonor. gDonor was synthesized via click chemistry and gel analysis confirms the synthesis of gDonor. (**c**) gDonor efficiently generates NHEJ in BFP-HEK cells after electroporation. The NHEJ frequency depends on the amount of gDonor. (**d**) gDonor with Cas9 can efficiently induce DNA cleavage and repair via HDR. BFP-HEK cells electroporated with gDonor/Cas9 had a similar

*Figure 4 continued on next page*

*Figure 4 continued*

HDR frequency as BFP-HEK cells electroporated with Cas9 RNP and donor DNA. (e) gDonor has a similar DNA cleavage pattern in cells as free gRNA and donor DNA (control). Deep sequencing analysis of BFP-HEK cells edited with gDonor/Cas9 and comparison to cells edited with Cas9 RNP and donor DNA (control). Cas9 with gDonor has an almost identical DNA cleavage profile as the unmodified control. The targeted Cas9 cleavage site for these experiments was at 64 locus (position of mutation), which is where most of the mutations were observed. (f) The gDonor/Cas9 complex was delivered into cells with cationic polymers, and the delivery efficiency was compared against cationic polymers complexed to unconjugated gRNA and donor DNA. gDonor/Cas9 complexed to PAsp(DET) was three times more efficient at generating HDR in BFP-HEK cells than PAsp(DET) complexed to Cas9 RNP and donor DNA. An additional control composed of a scrambled DNA conjugated to the gRNA did not increase the transfection efficiency of PAsp(DET). Student-t-test, significant difference from gDonor/Cas9, **$p<0.01$.

The following source data and figure supplements are available for figure 4:

**Source data 1.** Raw data from deep sequencing analysis.

**Figure supplement 1.** Synthesis and purification of gDonor.

**Figure supplement 2.** Mutation patterns made by gDonor and Cas9 are similar to that of free gRNA and Cas9.

**Figure supplement 3.** PAsp(DET) polymer nanoparticles complex both gDonor and Cas9.

**Figure supplement 4.** Dynamic light scattering analysis of PAsp(DET) complexes with Cas9 and gDonor.

gRNA. *Figure 4c* demonstrates that the gDonor was able to convert the BFP gene to the GFP gene via HDR with an efficiency similar to unmodified gRNA and Donor DNA, and thus both the gRNA and donor DNA of the gDonor are active. *Figure 4d* demonstrates that the NHEJ frequency induced by gDonor is dose dependent. In addition, deep sequencing analysis of the electroporated cells demonstrates that the gDonor cleaved its target sequence in cells with specificity and induced a similar pattern of indel mutations as unmodified gRNA control (*Figure 4e* and *Figure 4—figure supplement 2*). These results demonstrate that the gDonor can efficiently function as both a gRNA and a donor DNA.

## Delivery of gDonor into cells with polycations

We investigated if the gDonor could efficiently induce HDR in cells after delivery with cationic polymers. The cationic polymer, PAsp(DET), was selected as the initial polymer to deliver the gDonor because of its well established ability to deliver siRNA into cells and *in vivo* (*Miyata et al., 2008*; *Kim et al., 2010*, *2014*). The gDonor was mixed with Cas9 and complexed with PAsp(DET), and generated nanoparticles 150 nm in diameter that contained the Cas9-gDonor complex (*Figure 4—figure supplements 3* and *4*). The polymer nanoparticles were added to BFP-HEK cells and the HDR efficiency was determined by flow cytometry. *Figure 4f* demonstrates that gDonor significantly improves the ability of cationic polymers to simultaneously deliver Cas9, gRNA and donor DNA into cells. For example, the Cas9-gDonor complexed with PAsp(DET) induced an 8% HDR frequency in BFP-HEK cells, which was three times higher than that of the free gRNA and donor DNA complexed to PAsp(DET). Additional control cell experiments were conducted with a scrambled DNA conjugated gRNA, which had the same charge density as the gDonor. Cells were treated with the scrambled DNA-crRNA/Cas9 complexed with PAsp(DET) and a separate complex of donor DNA/PAsp(DET), and the HDR efficiency was measured. *Figure 4f* demonstrates that the scrambled DNA-crRNA conjugate did not improve the transfection efficiency of PAsp(DET), suggesting that the gDonor's ability to enhance the efficacy of PAsp(DET) is not related to stronger complexation. The gDonor represents a new reagent for improving the delivery of both Cas9 RNP and donor DNA into cells and has great potential for accelerating the development of Cas9 based therapeutics.

## Conclusions

In this report, we demonstrate that the gRNAs and donor DNA can be chemically modified at their terminal positions without losing activity. The tolerance of the donor DNA and gRNA to 5' modifications was exploited to develop a method for enriching cells that have a high chance of undergoing

HDR. In addition, we synthesized a gRNA-donor DNA conjugate (gDonor) that enabled the efficient delivery of Cas9 RNP and donor DNA into cells. We anticipate numerous applications of chemically modified gRNA and donor DNA for gene engineering given the wide variety of chemical modifications they tolerate.

## Materials and methods

### Materials

Unmodified crRNA, 5' Amine-crRNA, 5' Azide-crRNA, 5' Thiol-crRNA, 3' Amine-crRNA, 5' Amine-Donor, 3' Amine-Donor, 5' Azide-Donor, and various DNA sequences were purchased from Integrated DNA Technology (IDT). Phusion High-fidelity DNA Polymerase was purchased from NEB (Ipswich, MA). The Megascript T7 kit, the Megaclear kit, the PageBlue solution, the propidium iodide, and the PureLink genomic DNA kit were purchased from Thermo Fisher (Waltham, MA). Mini-PROTEAN TGX Gels (4–20%) were purchased from Bio-Rad (Hercules, CA). 4-(2-hydroxyethyl) piperazine-1-ethanesulfonate (HEPES) were purchased from Mandel Scientific (Guelph, ON). Sodium silicate was purchased from Sigma Aldrich (St. Louis, MO). Matrigel was purchased from BD Biosciences (San Jose, CA). DMEM media, non-essential amino acids, penicillin-streptomycin, DPBS and 0.05% trypsin were purchased from Life Technologies (Carlsbad, CA). EMD Millipore Amicon Ultra-4 100 kDa and 300 kDa were purchased from Millipore (Germany). *Streptococcus pyogenes* Cas9 (spCas9) and *Acidaminococcus sp. Cpf1* (AsCpf1) were purchased from the QB3 Macrolab from UC Berkeley. PAsp(DET) polymer was a generous gift from Dr. Kataoka's group (*Miyata et al., 2008*; *Kim et al., 2010*, *2014*).

### *In vitro* T7 transcription of tracrRNA ansd sgRNAs

TracrRNA and sgRNAs were synthesized using the *in vitro* transcription method with the MEGAscript T7 kit (Thermo Fisher) (*DeWitt et al., 2016*; *Richardson et al., 2016*). Purification of gRNAs was conducted using the MEGAclear kit, following the manufacturer's protocol.

### Synthesis of DBCO-crRNA

Dibenzocyclooctyne (DBCO)-crRNA was synthesized according to the synthetic scheme described in *Figure 1—figure supplement 1*. Amine-crRNA (either 5 or 3') (100 µM) was suspended in a 100 µL of DMSO and mixed with a 100-fold molar excess of Compound 1 (10mM). The reaction was incubated at room temperature for 16 hr and then purified with a desalting column (Micro Bio-Spin 30, Bio-rad). The concentration of the purified DBCO-crRNA was measured with a Nanodrop spectrophotometer. The synthesis of DBCO-crRNA was verified by conjugating it to either an azide modified DNA or azide-Cy5. DBCO-crRNAs (1 pmole) were incubated with a three times molar excess of 5' azide-DNA (3 pmole) or a 100 fold molar excess of azide-Cy5 (100 pmole) in 10 µL HEPES buffer to test the DBCO conjugation yield. The reaction mixture was run on a polyacrylamide gel and analyzed for crRNA content via densitometry or fluorescence intensity, and the reaction yield was determined by comparing the conjugate intensity to free DBCO-crRNA. The DBCO-crRNA samples with higher than 85% of reaction yield were used in further experiments.

### Synthesis of rhodamine-crRNA and Alexa 647-Donor DNA

NHS ester-Rhodamine and Alexa 647 NHS ester (Alexa647) were purchased from Thermo Fisher (Waltham, MA). One hundred times molar excess of NHS ester-Rhodamine or Alexa 647 (1 mM) were added to 5' amine-crRNA or 5' amine-DNA (10 µM) in pH 8.5 PBS (100 µL). After 4 hr of incubation at room temperature, a desalting column (Micro Bio-Spin 30, Bio-rad) was used to purify the conjugates. The conjugation yield was determined via fluorescence and samples with higher than an 85% reaction yield were used for further experiments.

### DNA-crRNAs synthesis and purification

The conjugation of donor DNA and crRNA (gDonor) was conducted using copper-free click chemistry (*Figure 1—figure supplement 2*). 5' Azide-DNA (86nt) was purchased from IDT and 5' Azide-DNAs with different lengths were synthesized from 5' amine-DNAs. 5' Azide-donor DNA (10 µM) was mixed with 5' DBCO-crRNA (10 µM) in DI water (50 µL). The solution was incubated at room

temperature overnight. The sample was analyzed via gel electrophoresis using a polyacrylamide gel (4–20% Mini-protean TGX Precast gel, Biorad). PAGE gel extraction was conducted to purify the gDonor conjugate. The DNA-crRNA band was cut with a sharp knife and eluted using the crush and soak method in nuclease-free water for 16 hr, and purified via ethanol precipitation additionally. 200 ng of crRNA, DNA, Azide-DNA, crRNA + DNA, and DNA-crRNA were analyzed via gel electrophoresis using a polyacrylamide gel (4–20% Mini-protean TGX Precast gel, Biorad) (*Figure 1—figure supplement 3*).

## BFP expressing cell culture

BFP-HEK293T cells and BFP-K562 cells were generated by infecting HEK293T and K562 cells with a BFP-containing lentivirus, followed by FACS-based enrichment, and clonal selection for cells expressing BFP with no silencing after 2–4 weeks (*Richardson et al., 2016*). The cells were STR profiled and mycoplasma contamination test result was negative. Cells were cultured in DMEM with 10% FBS, $1\times$ MEM non-essential amino acids, and 100 µg/mL Pen Strep. BFP-HEK293T cells were plated at a density of $10^5$ cells per well in a 24-well plate, one day before editing experiments were performed.

## Nucleofection

BFP-HEK cells were detached by 0.05% trypsin or gentle dissociation reagent, spun down at 600 g for 3 min, and washed with PBS. BFP-K562 cells were collected and washed with PBS. Nucleofection was conducted using an Amaxa 96-well Shuttle system following the manufacturer's protocol, using 10 µL of Cas9 RNP (Cas9 - 50 pmole, crRNA and TracrRNA - 60 pmole unless the amount is specified) (*Richardson et al., 2016*; *Lin et al., 2014*; *Schumann et al., 2015*). Nucleofection with Donor DNA had an additional 100 pmole of Donor DNA. After the nucleofection, 500 µL of growth media was added to the cells and they were incubated at 37°C in their tissue culture plates. The cell culture media was changed 16 hr after the nucleofection, and the cells were incubated until flow cytometry analysis was performed. The NHEJ frequency was quantified by determining the BFP negative population, and the HDR frequency was quantified by determining the GFP positive population. For the chemically modified gRNA activity tests, chemically modified crRNAs (30 pmole) were nucleofected with TracrRNA (30 pmole), and Cas9 (25 pmole) into BFP-HEK cells ($10^5$ cells).

## Chemically modified crRNA and Cpf1 nucleofection

AsCpf1 requires a TTTN PAM sequence instead of the NGG sequence of SpCas9 (*Zetsche et al., 2015*). We designed a crRNA for AsCpf1 that targets the BFP gene, and crRNA was purchased from IDT and additional modifications were conducted as described above. Nucleofection was conducted using the methods described above with crRNAs (30 pmole) and Cpf1 (25 pmole) into BFP-HEK cells ($10^5$ cells).

## Flow cytometry analysis and fluorescence microscopy

Flow cytometry was used to quantify the expression levels of BFP. The BFP-HEK and BFP-K562 cells were analyzed 5 days after Cas9 treatment. The BFP-HEK cells were washed with PBS and detached by trypsin. BFP and GFP expression were quantified using a BD LSR Fortessa X-20 and Guava easyCyte.

## Enrichment of HDR edited cells with fluorescent donor DNA

Alexa647 labeled 127 nt donor ssDNA was used for the enrichment experiments. BFP-HEK and BFP-K562 cells were nucleofected using an Amaxa 96-well Shuttle system following the manufacturer's protocol, using 10 µL of Cas9 RNP (Cas9 - 50 pmole, BFP sgRNA - 60 pmole, and 647-Donor - 50 pmole in $10^5$ cells). Cells were sorted using a BD influx cell sorter (BD Biosciences) in the Berkeley flow cytometry facility, 16 hr after the nucleofection. Positive gating captured the top 20% of Alexa647 positive cells, and negative gating captured the bottom 20% of cells, which had the least amount of Alexa647 fluorescence. The cells were cultured for an additional two to five days and then analyzed with flow cytometry. Fluorescence images were taken using a Zeiss inverted microscope and analyzed with Zen 2015 software (*Figure 3b* and *Figure 3—figure supplement 1*).

## Enrichment of mdx myoblast with gene editing

Primary myoblasts were isolated from C57BL/10ScSn-*Dmdmdx*/J (mdx) mice. The detailed procedure can be found in Rando et al. (*Rando and Blau, 1994*; *Conboy and Conboy, 2010*). Lipofection of Cas9 RNP and Alexa647 labeled 127nt donor ssDNA was conducted following previously published work (*Zuris et al., 2015*). Cells were sorted using a BD influx cell sorter (BD Biosciences) in the Berkeley flow cytometry facility, 16 hr after the nucleofection. Positive gating captured the top 20% of Alexa647 positive cells and negative gating captured the bottom 20% of cells, which had the least amount of Alexa647 fluorescence. DNA extraction was conducted 3 days later and PCR amplification of the mdx target region was conducted using the primer set (GAGAAACTTCTGTGATG TGAGGACATATAAAG and CAATATCTTTGAAGGACTCTGGGTAAAATATC). The HDR efficiency in cells was determined via restriction enzyme digestion of the PCR amplified target genes (*Fu et al., 2014*; *Lin et al., 2014*; *Schumann et al., 2015*). The donor DNA for these experiments contained a ClaI restriction enzyme site. The PCR amplicon of the mdx locus was incubated with 10 units of ClaI. After 16 hr of incubation at 37°C, the products were analyzed by gel electrophoresis using a 4–20% Mini-PROTEAN TGX Gel (Bio-rad) and stained with SYBR green (Thermo Fisher). Individual band intensity was quantified using ImageLab and the HDR efficiency was calculated using the following equation, where (a) is the uncleaved PCR amplicon and (b) and (c) are the cleavage products: $\frac{(b+c)}{(a+b+c)} x\ 100$.

## gDonor synthesis and purification

5' Azide-donor DNA was purchased from IDT. 5' Azide-donor DNA (10 µM) was mixed with 5' DBCO-crRNA (30 µM) in DI water (50 µL). The solution was incubated at room temperature overnight and the unreacted crRNA was removed by running the reaction solution through a 30k concentrator (Amicon Ultra, EMD Millipore). The gDonor reaction solution was analyzed via gel electrophoresis using a polyacrylamide gel (4–20% Mini-protean TGX Precast gel, Biorad) 200 ng of the reaction mixture was loaded into the gel. The yield of the gDonor was calculated by dividing the band intensity of the gDonor with the combined band intensities of the gDonor + unreacted Donor DNA. The intensity of the gDonor was multiplied by 8/13 to account for its higher molecular weight. PAGE gel extraction was conducted to purify the gDonor conjugate. The gDonor band was cut with a sharp knife and eluted using the crush and soak method in nuclease-free water for 16 hr, and isolated via ethanol precipitation. The purified gDonor was analyzed via gel electrophoresis using a polyacrylamide gel (4–20% Mini-protean TGX Precast gel, Biorad), 200 ng of crRNA, donor DNA, unpurified gDonor, and purified gDonor were loaded onto the gel.

## Deep sequencing analysis

The genomic region of the Cas9 target sequence was amplified by PCR using Phusion high-fidelity polymerase according to the manufacturer's protocol. Target genes were amplified first with primer sets (ATGGTGAGCAAGGGCGAGGAGC and TCGATGCCCTTCAGCTCGATGC). After PCR clean-up, the target gene was amplified with deep sequencing primers. The amplicons were purified using the ChargeSwitch PCR clean-up kit (Thermo Fisher) prior to the deep sequencing PCR. After barcoding and purification, the Berkeley Sequencing facility performed DNA quantification using a Qubit 2.0 Fluorometer (Life Technologies, Carlsbad, CA). A BioAnalyzer was then used for size analysis and qPCR quantification. The library was sequenced with the Illumina HiSeq2500 in the Vincent Coates Genomic Sequencing Laboratory at UC Berkeley. The analysis was conducted using the CRISPR Genome Analyzer (*Güell et al., 2014*).

Deep sequencing primers:
TCTTGTGGAAAGGACGAAACACCGTCATCTGCACCACCGGCAAGCT
TCTACTATTCTTTCCCCTGCACTGTCCGTCGTCCTTGAAGAAGATGGT

## Transfection of gDonor with PAsp(DET)

gDonor (5 µg in 10 µL), and TracrRNA (2 µg in 10 µL) were mixed in 80 µL of Cas9 buffer (50 mM Hepes (pH 7.5), 300 mM NaCl, 10% (vol/vol) glycerol, and 100 µM TCEP), and hybridized by incubating at 60°C for 5 min at RT for 10 min. Cas9 (8 µg in 10 µL) was added and incubated for 5 min at RT, and this solution was then added to the PAsp(DET) (10 µg in 20 µL) and incubated for 5 min at

RT to generate polymer nanoparticles. The particles were added to BFP-HEK cells ($10^5$ cells) at a Cas9 concentration of 16 μg/mL in 500 μL volume of culture medium for 16 hr. crRNA-TracrRNA/Cas9 + donor DNA were complexed with PAsp(DET) as a control and scrambled DNA-crRNA-TracrRNA/Cas9 and donor DNA were complexed with PAsp(DET) as a second control. Cell transfections with the two control nanoparticles were conducted following the same protocol used for transfecting cells with gDonor and TracRNA. Flow cytometry analysis was conducted 3 days after the nanoparticle treatment.

## Characterization of gDonor/PAsp(DET) nanoparticles

PAsp(DET) nanoparticles with gDonor and Cas9 were formulated as described above. The polymer nanoparticles were centrifuged at 17,000 g for 10 min, and the supernatant and pellet were collected. Each sample was mixed with a 100 μg of heparin for particle dissociation. The collected supernatant and pellets were run on a gel, and analyzed for the Cas9 and gDonor content in the polymer nanoparticles. Gel electrophoresis was performed using a 4–20% Mini-PROTEAN TGX Gel (Bio-rad) in Tris/SDS buffer, with a loading dye containing 5% beta-mercaptoethanol. PageBlue solution (Thermo Fisher) staining was conducted and imaged with ChemiDoc MP using ImageLab software (Bio-rad). For particle size measurements, a dynamic light scattering study was conducted using a Zetasizer Nano ZS instrument (Malvern Instruments Ltd., Worcestershire, UK) and a folded capillary cell (DTS 1060, Malvern Instruments). The reported particle size was measured 5 min after particle mixing.

## Statistics

All replicates were biological replicates. Statistical analysis was conducted using Prism7 software.

## Sequence information

Bold: T7 promoter sequence

### TracrRNA
**GTAATACGACTCACTATA**GGAACCATTCAAAACAGCATAGCAAGTTAAAATAAGGCTAGTCCG TTATCAACTTGAAAAAGTGGCACCGAGTCGGTGCT

### GFP sgRNA
**GTAATACGACTCACTATA**GCTGAAGCACTGCACGCCATGTTTTAGAGCTAGAAATAGCAAG TTAAAATAAGGCTAGTCCGTTATCAACTTGAAAAAGTGGCACCGAGTCGGTGCT

### Mdx sgRNA
**GTAATACGACTCACTATA**GGTCTTTGAAAGAGCAATAAGTTTTAGAGCTAGAAATAGCAAG TTAAAATAAGGCTAGTCCGTTATCAACTTGAAAAAGTGGCACCGAGTCGGTGCT

### GFP crRNA
GCTGAAGCACTGCACGCCATGTTTTAGAGCTATGCTGTTTTG

### GFP Donor sequence
127nt
GCCACCTACGGCAAGCTGACCCTGAAGTTCATCTGCACCACCGGCAAGCTGCCCGTGCCC TGGCCCACCCTCGTGACCACCCTGACGTACGGCGTGCAGTGCTTCAGCCGCTACCCCGACCACA TGA

86nt
CGGCAAGCTGCCCGTGCCCTGGCCCACCCTCGTGACCACCCTGACGTACGGCGTGCAGTGC TTCAGCCGCTACCCCGACCACATGA

### mdx Donor DNA sequence
TGATATGAATGAAACTCATCAAATATGCGTGTTAGTGTAAATGAACTTCTATTTAATTTTGAGGCTC
TGCAAAGTTCTTTgAAaGAGCAGCAGAATGGCTTCAACTATCTGAGTGACACTGTGAAGGAGA
TGGCCAAGAAAGCACCTTCAGAAATATGCCAGAAATATCTGTCAGAATTT

### Cpf1 crRNA sequence
TAATTTCTACTCTTGTAGATCGTCGCCGTCCAGCTCGACCAGGA

### Scrambled DNA
TTTGTTCAATCATTAAGAAGACAAAGGGTTTGTTGAACTTGACCTCGGGGGGGGATAGACATGGG
TATGGCCTCTAAAAACATGGCCA

## Acknowledgement

We thank Jennifer Doudna, Fuguo Jiang, Michael Conboy, Irina Conboy, Hyunjin Kim, Kazunori Kataoka, Hyo Min Park, Hunghao Chu, and Subhamoy Das, for advice and technical support. We thank Xiaojian Wang, Santanu Maity, and Corinne Sadlowski for chemistry advice and technical support.

## Additional information

### Competing interests
KL: Founder and employee of GenEdit. VAM: Employee of GenEdit. NM: Founder of GenEdit. The other authors declare that no competing interests exist.

### Funding
The authors declare that there was no funding for this work

### Author contributions
KL, Supervision, Investigation, Writing—original draft, Writing—review and editing; VAM, Conceptualization, Resources, Data curation, Formal analysis, Funding acquisition, Investigation, Visualization, Writing—original draft, Project administration, Writing—review and editing; AR, Resources, Data curation, Formal analysis, Validation, Visualization; ATC, MAD, Validation, Visualization, Methodology; JEC, NM, Resources, Methodology

### Author ORCIDs
Vanessa A Mackley, http://orcid.org/0000-0001-7734-2373
Jacob E Corn, http://orcid.org/0000-0002-7798-5309
Niren Murthy, http://orcid.org/0000-0002-7815-7337

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
