## [Decision Letter]

Thank you for submitting your article "Synthetically modified guide RNA and donor DNA are a versatile platform for CRISPR-Cas9 engineering" for consideration by *eLife*. Your article has been reviewed by two peer reviewers, and the evaluation has been overseen by Gordana Vunjak-Novakovic as the Reviewing Editor and Fiona Watt as the Senior Editor. The following individual involved in review of your submission has agreed to reveal their identity: Suzie Pun (Reviewer #3).

The reviewers have discussed the reviews with one another and the Reviewing Editor has drafted this decision to help you prepare a revised submission.

Summary:

The manuscript is a valuable systematic report of the extent to which terminal modifications are permitted on guide RNA and donor DNA in CRISIP/Cas9 and Cpf1 gene editing, with a focus on Cas9. The observation that guide RNA and donor DNA can be chemically conjugated is particularly important. In addition, the investigators apply fluorophore labeling of donor DNA as a means to enrich cells that have a higher probability of being genetically modified simply due to the amount of donor DNA associated with the cells. The results are valuable and expected to inform the design and use of bioconjugates with crRNA/Cpf1 guide RNA. In a broader sense, the study is an important advance in delivery technology.

Essential revisions:

We would like you to address the following concerns raised during the reviews, some of which will require the inclusion of additional experimental data.

1) You state: "Figure 2 demonstrates that the crRNA of AsCpf1 (from Acidaminococcus) tolerates chemical modification at its 3' end, in contrast 5' end modifications are less tolerated." However, the statistical significance is unclear (based on the large variance in the data), and only one 3' modification was investigated, which makes it difficult to draw a conclusion on a trend. Please revise the statement to the extent supported by data.

2) The use of fluorescent labeling enabled enrichment by a factor of 2. Please comment on the statistical significance of this enrichment and its practical importance with regard of simplifying the whole process.

3) In addition to histogram in Figure 3, the reader would benefit from the integral FACS data that will also illustrate gating used for sorting. Because the enrichment was not high, it is of interest to see if more stringent gating would have yielded a more pure cell population. We suggest that FACS data are included, along with the relevant discussion.

4) In the last paragraph of the subsection “A gRNA-donor DNA conjugate (gDonor) can function as a gRNA and a donor DNA”, the authors state that gDonor was synthesized with a 40% yield. Please specify how this was calculated since Figure 4 seems to show much lower yield of conjugated gDonor (third lane).

5) The statement that gDonor increases PAsp(DET)-mediated HDR is not supported rigorously enough, because of co-delivery of donor DNA and gRNA. It is possible that the longer gDonor results in more efficient packaging of Cas9. To be certain, one more control that should be included: gRNA/Cas9 + Donor DNA with extra nucleic acids at the 5' gRNA to increase its' polyanionic character for packaging.

---

## [Author Response]

*Essential revisions:*

*We would like you to address the following concerns raised during the reviews, some of which will require the inclusion of additional experimental data.*

*1) You state: "Figure 2 demonstrates that the crRNA of AsCpf1 (from Adisaminococcus) tolerates chemical modification at its 3' end, in contrast 5' end modifications are less tolerated." However, the statistical significance is unclear (based on the large variance in the data), and only one 3' modification was investigated, which makes it difficult to draw a conclusion on a trend. Please revise the statement to the extent supported by data.*

We thank the reviewers for their thoughtful comment, and have performed additional experiments to investigate if the Cpf1 gRNA is more tolerant to 3’ modifications than 5’ modifications. We purchased or synthesized Cpf1 gRNAs with amine, DBCO, and azide modifications, at either the 3’ or 5’ end, and investigated their ability to generate NHEJ in BFP-HEK cells. These new results are presented in Figure 2 and demonstrate that the Cpf1 gRNA is more tolerant to modifications at its 3’ end than its 5’ end. Cpf1 gRNAs with 5’ modifications composed of either an azide or DBCO had a statistically significant reduction in NHEJ efficiency in comparison to gRNAs that had 3’ amine or azide modifications. We have rewritten the text to describe this additional experiment and also to more accurately reflect the experimental data. It now states: “Figure 2 demonstrates that the crRNA from AsCpf1 (from Acidaminococcus) tolerates chemical modifications such as amine, azide, and DBCO at its 3’ end; in contrast, bulky 5’ end modifications such as azide and DBCO are less tolerated.[…] In contrast, BFP-HEK cells electroporated with crRNAs with 5’ modifications and Cpf1 had a reduction in NHEJ frequency and generated only 50-80% of the NHEJ levels as cells treated with unmodified crRNA.”

*2) The use of fluorescent labeling enabled enrichment by a factor of 2. Please comment on the statistical significance of this enrichment and its practical importance with regard of simplifying the whole process.*

We performed the cell enrichment experiments presented in Figure 3 in triplicate. For each experimental group, three separate cell culture plates were treated with Cas9 RNP and fluorescent donor DNA, sorted and analyzed for HDR. We have performed a one-way ANOVA, post-hoc Tukey test between the average HDR frequency of Alexa647 enriched samples versus the average HDR frequency of control unsorted cells, and obtained a p value lower than 0.01, demonstrating statistical significance.

Enriching cells for gene editing, non-invasively, via FACS, has the potential to significantly accelerate gene editing cell culture protocols, because it will reduce the number of clones that need to be isolated and sequenced by a factor of 2. The identification of edited cells now requires isolating clones, growing them up and performing deep sequencing. The flow sorting technique presented here, requires minimal additional cost, and will reduce the number of clones that need to be isolated by a factor of 2, and should therefore lower the cost of isolating edited cells. We anticipate that flow sorting based on the donor DNA fluorescence will be particularly useful for editing primary cells, because these cells cannot be easily cultured ex vivo and have a low gene editing efficiency. We have added additional text to the manuscript to explain the potential advantages of flow sorting based on the donor DNA fluorescence, it now states: **“**Enriching cells for gene editing, non-invasively via FACS, has the potential to significantly accelerate gene editing cell culture protocols, because it will reduce the number of clones that need to be isolated and sequenced”.

*3) In addition to histogram in Figure 3, the reader would benefit from the integral FACS data that will also illustrate gating used for sorting. Because the enrichment was not high, it is of interest to see if more stringent gating would have yielded a more pure cell population. We suggest that FACS data are included, along with the relevant discussion.*

We thank the reviewer for the thoughtful comment. We have now included the original FACS and flow cytometry data for the experiment described in Figure 3, to show the gating used for sorting and quantification. The top image of Figure 3 shows the histogram of Alexa647 fluorescence and the gating that was used to delineate Alexa647-negative and Alexa647-positive cells. The fluorescent microscopy images of the Alexa647-negative and Alexa647-positive cells are also shown in Figure 3.

We have performed an additional experiment to determine if using a more stringent gate for cells that internalized donor DNA would lead to cell populations with higher levels of HDR. We therefore performed an HDR experiment on BFP-HEK cells, transfected with fluorescently labeled donor DNA and Cas9 RNP, and isolated the top 10% of Alexa 647+ cells and the top 20% of Alexa 647+ cells, via flow sorting, and compared their HDR efficiency. We did not observe a difference in HDR efficiency between these two cell populations, suggesting that a 2 fold increase in HDR efficiency is the maximum enrichment we can generate with flow sorting based on the donor DNA fluorescence.

*4) In the last paragraph of the subsection “A gRNA-donor DNA conjugate (gDonor) can function as a gRNA and a donor DNA”, the authors state that gDonor was synthesized with a 40% yield. Please specify how this was calculated since Figure 4 seems to show much lower yield of conjugated gDonor (third lane).*

We thank the reviewers for pointing out this discrepancy. The yield in the gDonor synthesis experiment was calculated by running the gDonor reaction mixture on a gel and quantifying the band intensities of the gDonor and the unreacted Donor DNA. The yield of the gDonor was determined by dividing the band intensity of the gDonor with the combined band intensities of the gDonor + unreacted Donor DNA, which represents the maximum possible yield. The intensity of the gDonor was multiplied by 8/13 to account for its higher molecular weight.

The gel in Figure 4 is difficult to interpret because the crRNA was used in a large excess and the intensity of the donor DNA and gDonor bands therefore appear small. We have performed another synthesis of the gDonor and have removed the excess crRNA in the reaction via a spin filter. The resulting gDonor reaction was run on a new gel and quantified as described above. This new gel is shown in Figure 4, and the band intensities are now clearly visible and correspond to a 40% yield. We have described the procedure used to calculate the gDonor synthesis yield in the Methods.

*5) The statement that gDonor increases PAsp(DET)-mediated HDR is not supported rigorously enough, because of co-delivery of donor DNA and gRNA. It is possible that the longer gDonor results in more efficient packaging of Cas9. To be certain, one more control that should be included: gRNA/Cas9 + Donor DNA with extra nucleic acids at the 5' gRNA to increase its' polyanionic character for packaging.*

We thank the reviewer for their thoughtful comment. As suggested by the reviewers, we have performed additional transfections experiments with gRNA that has extra nucleic acids at its 5’ end, and have compared its transfection efficiency against gDonor. This new data is described in Figure 4, and demonstrates that increasing the gRNA with nucleic acids does not increase transfection efficiency. This new data is described in the text as follows: “Additional control cell experiments were conducted with a scrambled DNA conjugated gRNA, which had the same charge density as the gDonor. […] The gDonor represents a new reagent for improving the delivery of both Cas9 RNP and donor DNA into cells and has great potential for accelerating the development of Cas9 based therapeutics.”